# Pore Formation Mechanism of A-Beta Peptide on the Fluid Membrane: A Combined Coarse-Grained and All-Atomic Model

**DOI:** 10.3390/molecules27123924

**Published:** 2022-06-18

**Authors:** Yuxi Dai, Zhexing Xie, Lijun Liang

**Affiliations:** 1College of Automation, Hangzhou Dianzi University, Hangzhou 310018, China; 20068815@hdu.edu.cn; 2College of Accounting, Hangzhou Dianzi University, Hangzhou 310018, China; 20140935@hdu.edu.cn

**Keywords:** pore formation, multiscale modeling, Alzheimer’s disease, ionic channel

## Abstract

In Alzheimer’s disease, ion permeability through the ionic channel formed by Aβ peptides on cellular membranes appears to underlie neuronal cell death. An understanding of the formation mechanism of the toxic ionic channel by Aβ peptides is very important, but remains unclear. Our simulation results demonstrated the dynamics and mechanism of channel formation by Aβ1-28 peptides on the DPPC and POPC membrane by the coarse-grained method. The ionic channel formation is driven by the gyration of the radius and solvent accessible molecular surface area of Aβ1-28 peptides. The ionic channel formation mechanism was explored by the free energy profile based on the distribution of the gyration of the radius and solvent accessible molecular surface area of Aβ1-28 peptides on the fluid membrane. The stability and water permeability of the ionic channel formed by Aβ peptides was investigated by all-atomic model simulation. Our simulation showed that the ionic channel formed by Aβ1-28 peptides is very stable and has a good water permeability. This could help us to understand the pore formation mechanism by Aβ1-28 peptides on the fluidic membrane. It also provides us with a guideline by which to understand the toxicity of Aβ1-28 peptides’ pores to the cell.

## 1. Introduction

Alzheimer’s disease (AD) is a progressive, irreversible neurological disease [1,2,3]. It is the result of protein misfolding which transforms the three-dimensional conformations from native to nonnative (insoluble) folded structures [4,5,6]. One increasingly accepted hypothesis is that oligomer amyloid seeds are the toxic species [7,8,9,10]. The mechanism of toxicity involves the penetration of small oligomers into the membrane and formation of unregulated ion channels, which lead to ion leakage and, ultimately, cell death [11,12,13,14,15]. Scala et al. found that calcium could pretreat through the pore formed by beta-amyloid peptide [16]. However, the hypothesis of an ionic channel formed from Aβ peptides is still under the debated. In addition, given their dynamic nature, the experimental structures for these channels are still very difficult to obtain [17,18]. Standard tools of structural biology have failed to provide the high-resolution 3D structures of the oligomers of the Aβ peptide, especially on the membrane. Only low-resolution structural data from circular dichroism, electron microscopy and atomic force microscopy measurements are available [19,20,21]. In the groundbreaking 2020 study by Natàlia Carulla and coworkers, the first atomic structures of Aβ(1–42) oligomers were resolved [22]. These elegant findings indicate that membrane-embedded Aβ(1–42) oligomers form edge-conductivity pores [22]. However, a comprehensive understanding of the ion channels formed by Aβ peptides is still unclear, and thus, these need to be explored. 

To understand the formed ionic channel in much greater depth, theoretical calculations, especially molecular dynamics (MD) simulations, have been used to explore the A-beta aggregation process [23,24,25,26,27]. Nussinov and his coworkers designed Aβ channels based on the solid-state NMR-based β strands, and they found that the ion-permeable Aβ channels are consistent with the results from electron microscopy [28]. Mustata et al. found that small peptide fragments could combine to form an ionic channel [29]. These theoretical works greatly enhanced our understanding of the property of the Aβ ionic channel. However, all these works used the designed Aβ ionic channel as the initial structure to study the shape and ion permeation of the ionic channel. The dynamics and mechanism of Aβ ionic channel formation on the fluidic membrane are still unclear. Understanding the formation process of the Aβ ionic channel on the membrane is very important. It represents not only a fundamental area for academic research, but also gives us an enormous opportunity to improve the quality of life of AD patients. All-atomic (AA) simulation could provide the atomic details and dynamics of the aggregation process of the molecules; however, the timescale is limited up to hundreds of nanoseconds for a large system. Based on Zhou’s work [30], coarse grained (CG) MD simulation combined with the MARTINI V2.1 force field could model Aβ core peptides well, and it is widely used and validated as an effective model for studying the assembly of short peptides [31,32,33]. To better understand the Aβ channel formation process, CGMD simulation was used to perform the system with 25 Aβ1-28 peptides on the membrane. To make up for the shortcomings of insufficient atomic detail in CGMD, the combination of AA and CG was used to explore the formation process of the Aβ ionic channel. The combined AA and CG method has been successfully used to explore peptide aggregation and pore formation by alamethicin peptides in a hydrated lipid bilayer [34]. Our simulation showed that the ionic channel formed by Aβ1-28 peptides on the fluid lipid bilayer membrane has a good water permeability. 

## 2. Computational Details

### 2.1. Coarse Grained Simulation

The soluble Aβ1-28 peptide (PDB ID: 1AMB) [35], defined as Aβ peptide, was equilibrated for 10 ns in a water box. The equilibrated Aβ peptides were then transferred into a CG model based on the MARTINI V2.1 force field [36]. After that, 25 Aβ peptides, with the distance of center of mass (COM) between each other of 4.0 nm, were produced in the *x–y* plane. All CG Aβ peptides were inserted into a 1,2-dipalmitoyl-sn-glycero-3-phosphocholine (DPPC) membrane, and water molecules were placed in both sides of the membrane. The system consists of 25 Aβ peptides, 1000 DPPC lipids and 20705 water beads in a box of 20 nm × 20 nm × 11.0 nm. The parameters of Aβ peptides, DPPC lipids and water beads were extracted as described in MARTINI V2.1 force field. One bead represents the main chain of one residue in Aβ peptides, the benzene ring of phenylalanine is represented by three beads and one water bead represents four water molecules. To investigate the effect of the lipid membranes on the formation process of the ionic channel, another system including 1-palmitoyl-2-oleoyl-sn-glycero-3-phosphocholine (POPC) lipids was also constructed. Two systems, named as AMB-DPPC and AMB-POPC, were equilibrated to 100 ns equilibration by GROMACS 4.6.5 software [37]. Then, all simulations were performed for 5 μs with the time step of 20 fs. *NpT* ensembles were used with the constant pressures of 1 atm and constant temperature of 310 K. The cutoff for the non-bonded van der Waals interaction was set at 1.2 nm.

### 2.2. All-Atom Molecular Dynamics in an Explicit Solvent

After 5μs CG simulation, the system, including 25 Aβ peptides, DPPC lipids and water beads, was transferred into the AA model with a CHARMM 36 force field [38] by back-mapping tool [39]. The whole system included ca. 420,000 atoms in the AA model. The GROMACS 4.6.5 software was used to perform the simulation. All atoms, including hydrogen atoms, were represented explicitly with the time step of 2 fs, and the length of bonds between hydrogen atoms and other atoms was constrained using SHAKE algorithm [40]. The cutoff for the non-bonded van der Waals interaction was set by a switching function starting at 1.0 nm and reaching zero at 1.2 nm. In all MD simulations, the particle mesh Ewald (PME) summation [41] was used to calculate the long-ranged electrostatic interactions with periodic boundary conditions (PBC), with a cutoff distance of 1.2 nm for the separation of the direct and reciprocal space. All MD simulations were carried out in NpT ensemble with the constant pressures of 1 atm and constant temperature of 310 K, respectively.

## 3. Results and Discussion 

The combination of CG and AA simulations allowed for a complete description of peptides and the pore formation. In the CG simulation process, the pore formation from the Aβ peptides’ aggregation was observed. In addition, the transition process from the fibril-type aggregation to pore formation was observed in CG simulation. The AA simulation shows penetration of water molecules and ions into the lipid bilayer through the ionic pore formed by Aβ peptides. The results are discussed in detail in the following subsections.

### 3.1. Spontaneously Aggregation of Aβ Peptides

All the Aβ peptides remain in the membrane-bound state in the DPPC membrane and laterally diffuse to assemble in aggregates that slowly grow in size. The number of clusters of peptides was measured, with the condition that the two peptides belonged to the same cluster if the distance between the COM of these two peptides was less than 1.5 nm. The evolution of the number of peptide clusters as a function of simulation time is shown in Figure 1. The number of clusters rapidly decreased to 6 in less than 200 ns CG simulation. This shows us that the Aβ peptides tend to aggregate into clusters on the membrane. After a 500 ns simulation, the number of clusters remained at 4 until the end of the 5 μs simulation. The simulation was repeated three times, and the number of clusters remained at three and three in the other two trajectories. To explain the aggregation process more clearly, as seen in Figure 2, the snapshots of peptide clusters were captured during the simulation of AMB-DPPC system. Water molecules and DPPC lipids were not presented in the snapshot for clarity. At time = 100 ns, the Aβ peptides had aggregated into eight clusters. Therein, the largest cluster had six Aβ peptides, and all of them had fiber-like parallel aggregation. This shows us that the Aβ peptides are prone to aggregate into fiber even in the membrane. After that, at time = 400 ns, 25 Aβ peptides had aggregated into 4 clusters. The biggest cluster had seven Aβ peptides; the shape of this cluster had changed from the fiber-like to pore-like arrangement, and the other three clusters were not pore shaped. At 4000 ns, the number of cluster was still four, and the number of pore shaped clusters increased to three. As seen in Figure 2D, three clusters aggregated by Aβ peptides were pore shaped. This shows us that the Aβ peptides tend to form the pore shape in the DPPC membrane. To investigate the aggregation process more deeply, the solvent-accessible surface area (SASA) of Aβ peptides in the DPPC membrane was calculated. The correlations between Aβ peptides and the DPPC membrane is provided by examining SASA and the radius of gyration (Rg) of Aβ peptides [42]. The free energy landscape is determined by calculating the normalized probability from a possibility distribution of SASA and Rg, as seen in Equations (1) and (2) [43], where *X* and *Y* represent the SASA and Rg.


(1)
$$P(X,Y)=\frac{1}{Z}\exp[-\beta W(X,Y)]$$



(2)
$$\Delta G=W(X_{2},Y_{2})-W(X_{1},Y_{1})=-RT\log[\frac{P(X_{2},Y_{2})}{P(X_{1},Y_{1})}]$$


As shown in Figure 3, the free energy contour maps with two reaction coordinations, SASA and Rg, are measured. Two important structures could be exacted from the local minimum. One stable structure is the fiber-like structure, and the most stable one is the pore-shaped structure. As mentioned in Yu’s paper [42], the change in structure is accompanied by the change in SASA and Rg. It could greatly alter the free energy with the change in enthalpy and entropy, and the Aβ peptides could aggregate into a pore-shaped structure or fiber-like structure as seen in Appendix A. The value of SASA and Rg is 427.0 nm^2^ and 3.60 nm for the pore-like structure, respectively, and it is 428.0 nm^2^ and 4.12 nm for the fiber-like structure, respectively. This shows that the value of SASA for these two structures is almost the same. The energy barrier between these two structures is ca. 2.5 kcal/mol from the fiber-like structure to the pore-shaped structure, and it is ca. 3.7 kcal/mol from the pore-shaped structure state to the fiber-like structure state. Although the pore-shaped structure for aggregation of Aβ peptides is much more stable, the energy barrier between these two structures is not very high. This shows us that the conformation of aggregated Aβ peptides from a pore-shaped structure and a fiber-like structure is not very difficult.

### 3.2. Effect of Lipid Membrane

To investigate the effect of the lipid membrane on Aβ peptides’ aggregation, the POPC membrane was used to replace the DPPC membrane in the Aβ peptides aggregation process. In the system including Aβ peptides, DPPC lipids and water molecules, we found only 15 Aβ peptides still remained in the membrane-bound state, and the other 10 Aβ peptides remained in a surface-bound state after the 5 μs simulation. As seen in Figure 1, the number of clusters rapidly decreased to 1, and the 15 Aβ peptides remaining in the POPC membrane-spanning state also aggregated into a pore formation structure such as that on the DPPC membrane. This shows us that Aβ peptides tend to aggregate into a pore-like structure in the POPC membrane. Compared with the Aβ peptides’ pore-shaped structure on the DPPC structure, the pores formed by Aβ peptides on the POPC membrane are smaller. The shape of the pore formed by Aβ peptides on the POPC membrane is more irregular than that on the DPPC membrane. This implies that it is more difficult for the water molecules and ions to pass through the pore formed on the DPPC membrane than that on the POPC membrane. It shows us that the types of lipid membrane could affect the shape and properties of the pore formed by Aβ peptides. Although 10 Aβ peptides are excluded by POPC membrane, it interacts with the remaining 15 Aβ peptides in the membrane and combines them into one cluster. This also shows us that the interaction between different Aβ peptides’ monomers is very strong. In addition, to investigate the shape of Aβ channels more deeply, the order parameter (S) of the largest Aβ peptides’ cluster in two systems on different membrane was measured as a function of simulation time. The alpha C atoms of the Aβ channel in the largest cluster were used for the analysis. As shown in Figure 4, the S value of the Aβ channel on two different membranes decreased in the first stage with the aggregation of the Aβ peptides. However, the S value of the Aβ channel on the DPPC membrane is much higher than that of the Aβ channel on the POPC membrane. This means that the arrangement of Aβ peptides in the Aβ channel on the DPPC membrane is more parallel than that on the POPC membrane. It is also the reason why the pore formed by Aβ peptides on the DPPC membrane is larger than that formed by Aβ peptides on the POPC membrane. Due to the membrane fluidity, the S value of the Aβ channels from two systems fluctuated widely in the simulation. It facilitated the conformation change in Aβ peptides during the aggregation process.

### 3.3. Water Permeation of the Pore

Monitoring the penetration of water molecules into the holes and cavities formed by Aβ peptides’ aggregates is a very effective way to assess pore formation in a membrane patch in MD simulations. It is also important to evaluate the toxicity of pores formed by Aβ peptides. The pore formed on the DPPC membrane was selected to investigate the permeation of water molecules. During the CG simulations, only very few water beads interact with the Aβ channel inside the lipid bilayer. In the CG model, one single water bead represents four water molecules, for simplicity. Thus, the total number of water molecules greatly decreases, leading to insufficient visits of water to the channel. Compared to 0.28 nm for an AA water molecule, the vdW diameter of a single water molecule bead is 0.5 nm, which is much larger than the water molecules in the AA model. It is much more difficult for the water bead to enter into the Aβ channel in the membrane. The AA simulation was performed by 100 ns to explore the permeation details of the Aβ channels. The initial structure of the AA model is back mapped from the last structure in the CG simulation, as in our precious work [44]. 

As shown in Figure 5A, the water molecules could not interact with the inner residue of the Aβ channel in the initial structure. In the AA simulation, the pore was much better established. After performing the 25 ns AA simulation, water molecules could pass through the Aβ channels, as shown in Figure 5B. After that, the conformation of the Aβ channels changes a little in the remainder of the simulation. This shows us that the Aβ channel is very stable on the DPPC membrane, and it confirmed the mechanism of the Aβ channel’s formation on the membrane. Limited by the simulation time, the pore of the Aβ channel from the top view is not so cyclical; the radius is estimated ca. 0.55 nm based on the area of the pore. In Nussinov’s simulation, the diameter of the Aβ channel is from 0.6 nm to 3.7 nm, from 12 mer to 36 mer peptides [28]. Since the channel is aggregated by seven Aβ peptides in our simulation, the pore size is a little smaller than that from Nussinov’s simulation. To investigate the Aβ channel in depth, the passage and distribution of water molecules in the channel were analyzed. Most of water molecules in the Aβ channels could interact with the residues of the Aβ channels directly (the distance between water oxygen atoms and the atom of the peptide is less than 0.4 nm), and the remaining water molecules participated in the water networks to assist with the peptide–water interaction. The water molecule diffusion in Aβ channels is very slow in the initial state, and it is much faster after the expansion of Aβ channels after 25 ns. The density of water molecules and Na^+^ ions in the Aβ channels along the *z* axis was calculated; therein, the *z* axis of the Aβ channel is from 2.32 nm to 6.55 nm. From the data in Figure 6, the density of water molecules in the Aβ channel is really low compared with that of the bulk. The density of Na^+^ ion near the tail of membrane is really high due to the negative charge of the phosphate group. On the basis of the density of water molecules and Na^+^ ions along the *z* axis of the pore, the potential of mean force (PMF) was estimated as follows: 


$$\Delta G_{\mathrm{PMF}}=-k_{B}T\ln(\rho_{z}/\rho_{\mathrm{bulk}})$$


Herein, *k_b_* is the Boltzmann constant, *T* is the simulation temperature, $\rho_{z}$ is the density of water or Na^+^ in the *z* axis of pore and $\rho_{\mathrm{bulk}}$ is the density of water or Na^+^ in the bulk region. An accurate equilibrium PMF relevant to ion permeation should be obtained from the free energy calculations with the umbrella sampling method. However, the ion-density-based PMF could still provide us with the rough relative free energy changes for solvents. The energy barrier for the water molecule to pass through the Aβ channel is ca. 2.51 kcal/mol, and it is 2.02 kcal/mol for the Na^+^ ion. This is a very low energy barrier for the water molecule and Na^+^ ion to pass through the Aβ channel. 

**Figure 5 molecules-27-03924-f005:**
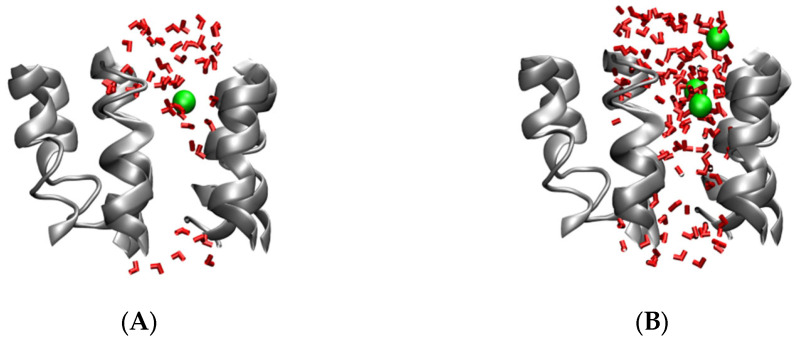
(**A**) is the initial state of the all atomic simulations from a side view; (**B**) the state after 100 ns, all atomic simulations in side view. The water molecules within 0.8 nm of the channel are shown as a red bonds model, Na^+^ ions within 0.8 nm of the channel are shown in a green CPK model and the Aβ channel is shown by cartoon model in VMD.

**Figure 6 molecules-27-03924-f006:**
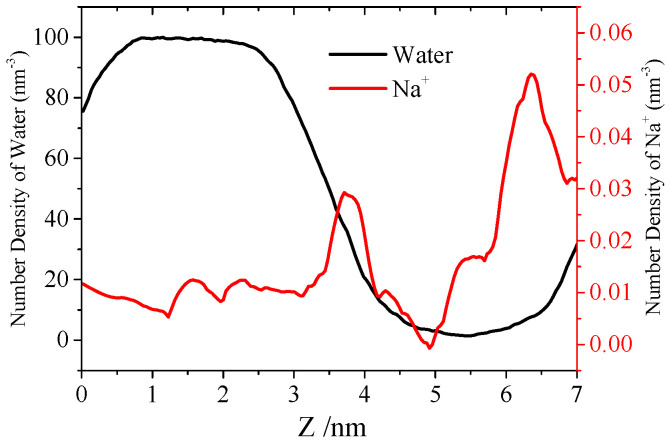
The density of water molecules (black line) and Na^+^ ions (red line) along the *z* axis of the DPPC membrane.

## 4. Conclusions

In summary, the pore formation mechanism of Aβ peptides on the membrane and the pore permeation were investigated by combined CG and AA simulation. In the CG simulation, pore-like and fiber-like structures in the aggregation of Aβ peptides on the membrane were observed. The free energy landscape was measured as a function of the change in Rg and SASA, and the pore-like structure is more stable than the fiber-like structure on the membrane. The aggregates do not consist of a specific number of peptides on the membrane, but rather they grow in size over time. In addition, the membrane lipid composition also has an effect on the aggregation process of Aβ peptides. Herein, the pore formation of Aβ peptides on the membrane was observed, and water permeation of the Aβ channel was investigated by AA simulation. Our results demonstrate that the Aβ channel formed by Aβ peptides on the DPPC membrane could have good water permeation.

## Figures and Tables

**Figure 1 molecules-27-03924-f001:**
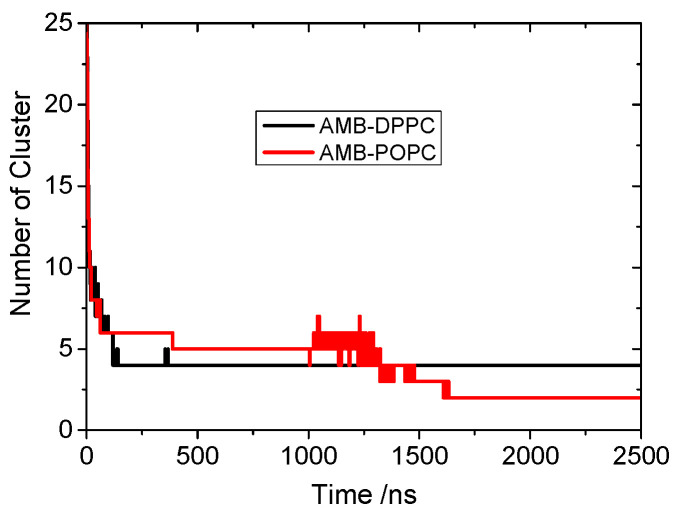
The number of peptide (AMB) clusters as a function of simulation time in different membrane systems: DPPC (black line) and POPC (red line).

**Figure 2 molecules-27-03924-f002:**
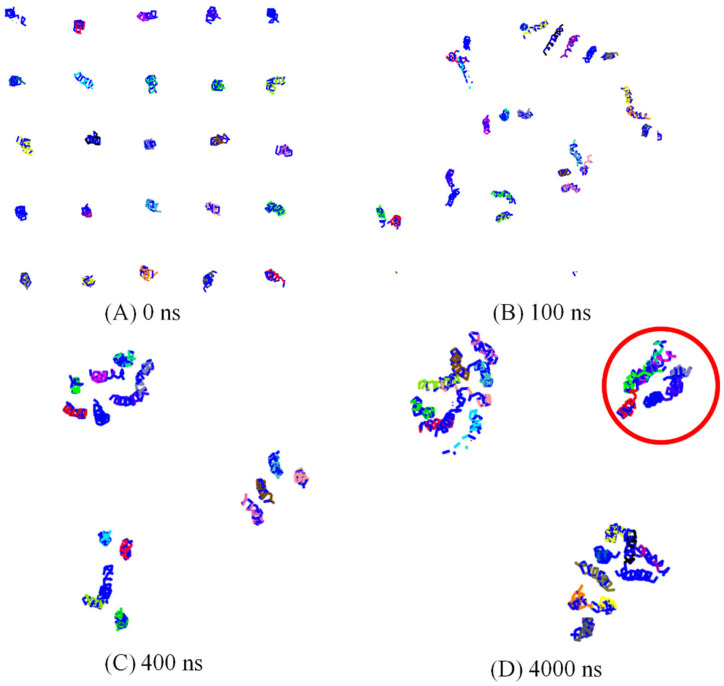
The snapshots of peptides clusters were captured in the simulation of the AMB–DPPC system along simulation time: (**A**) 0 ns; (**B**) 100 ns; (**C**) 400 ns and (**D**) 4000 ns. Water molecules and DPPC lipids were not presented in the snapshot for clarity.

**Figure 3 molecules-27-03924-f003:**
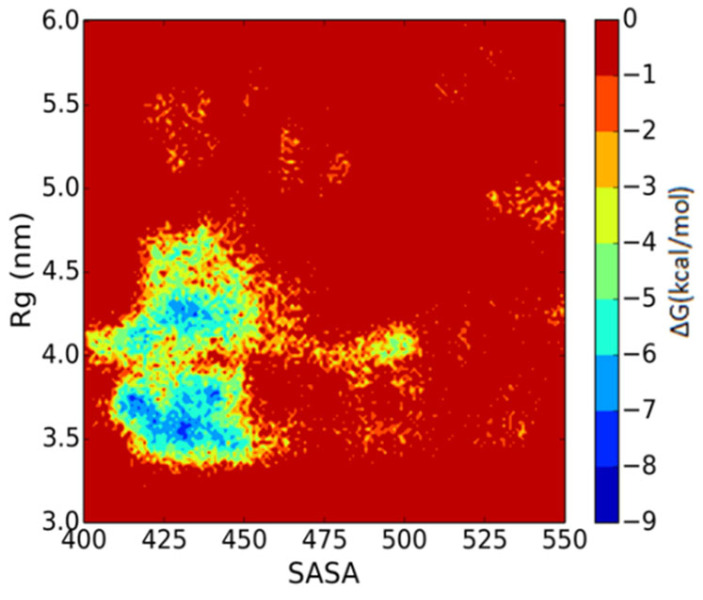
Free energy maps vs. two reaction coordinates, the Rg and SASA.

**Figure 4 molecules-27-03924-f004:**
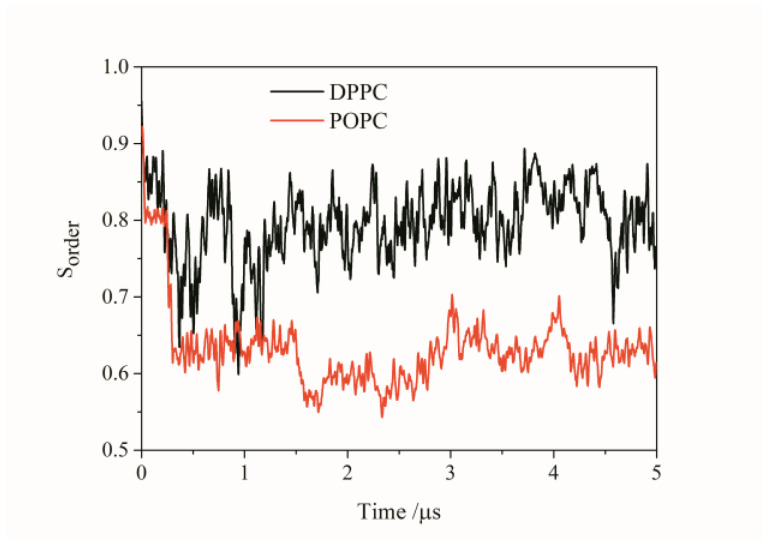
The order parameter of the alpha C atoms of the largest Aβ channel as a function of time on a two lipids membrane: POPC and DPPC.

## Data Availability

Not applicable.

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
