# Peer review of "Pore Formation Mechanism of A-Beta Peptide on the Fluid Membrane: A Combined Coarse-Grained and All-Atomic Model"

_molecules, 2022, doi:10.3390/molecules27123924_

Round 1

Reviewer 1 Report

In this manuscript, the authors present a study concerning the aggregation of AB peptides in the DPPC membrane in contact with water molecules on both sides. The results of this system are compared with those concerning the POPC membrane, for which pores are smaller and have more irregularshape. Coarse grained (CG) MD simulations are used as a first approach, and after a simulation of 5000 ns the system modeled by “beads” was mapped into the all-atomic description of the system in order to study water and sodium permeation of the pore. The standard simulation codes , GROMACS and CHARMM, were used for simulations as well as the gyration radius, SASA and PMF are quantities generally used in the literature of peptide and protein assembly. From CG simulations, the authors found that the pore-structure is more stable than the fiber structure, although the transition barrier between the two structures is not too much higher than that of the reverse process. Examination of the pore-structure with AA simulation finally demonstrated that both water molecules and sodium ion permeate the pore with a low barrier.

I have found this article not very clear and therefore I disagree with publication in Molecules in this form. Major revision required. Authors should clarify the following points.

(1) Figure (1) seems inconsistent with the definition of cluster given in the text because it includes the starting configuration of 25 peptides.

(2) In Figure (3) it is difficult to identify the two structures (pore and fiber) mentioned in the text.

Can the authors provide images of these structures and the corresponding gyration radius and SASA?,In the text, authors should specify the solvent (water?) for which SASA was computed. Based on the results in Figure (3) the authors state that the gyration radius and the SASA are decisive in the aggregation process. Actually, the language used (“driven forces”) appears not correct, and in the text there is no clear interpretation of the two quantities. Furthermore, the two quantities have been used in the literature (see for example Yu and Schatz J. Phys. Chem. B 2013, 117, 9004-9013) and I think that adding a few references on this topic can help the reader understand the relevance of the two quantities in the aggregation process.

(3) Please indicate the unit for density in Figure 6.

(4) Conclusion should be improved. What is the relevance of conclusions drawn from only one structure of the pore?

Author Response

Reviewer: 1

Recommendation: I have found this article not very clear and therefore I disagree with publication in Molecules in this form. Major revision required. Authors should clarify the following points.

Comments:

In this manuscript, the authors present a study concerning the aggregation of AB peptides in the DPPC membrane in contact with water molecules on both sides. The results of this system are compared with those concerning the POPC membrane, for which pores are smaller and have more irregularshape. Coarse grained (CG) MD simulations are used as a first approach, and after a simulation of 5000 ns the system modeled by “beads” was mapped into the all-atomic description of the system in order to study water and sodium permeation of the pore. The standard simulation codes , GROMACS and CHARMM, were used for simulations as well as the gyration radius, SASA and PMF are quantities generally used in the literature of peptide and protein assembly. From CG simulations, the authors found that the pore-structure is more stable than the fiber structure, although the transition barrier between the two structures is not too much higher than that of the reverse process. Examination of the pore-structure with AA simulation finally demonstrated that both water molecules and sodium ion permeate the pore with a low barrier.

Author reply: We sincerely thank the reviewer’s positive and constructive suggestions which strengthened our study.

  • Figure (1) seems inconsistent with the definition of cluster given in the text because it includes the starting configuration of 25 peptides.

Author reply: Thanks. We should explain the definition of cluster more clearly. In simulation, the 25 Aβ peptides with the distance of center of mass (COM) between each other is 4.0nm in initial structure. The definition of one cluster is that the distance between COM of peptides is less than 1.5nm, and the number of cluster of peptides were measured. Thus, the number of cluster is 25 for the initial structure of system.

  • In Figure (3) it is difficult to identify the two structures (pore and fiber) mentioned in the text. Can the authors provide images of these structures and the corresponding gyration radius and SASA?

Author reply: Thanks for the nice suggestion. As seen in Figure S1, two structure of Aβ peptides aggregation could be divided into fiber-like structure and pore-like structure. The value of SASA and Rg is 427.0 nm2 and 3.60nm for pore-like structure, and it is 428.0 nm2 and 4.12nm for fiber-like structure, respectively. It shows the value of SASA for these two structure is almost the same. We have revised this part in the manuscript.

Figure S1. The structure of Aβ peptides aggregation: (A) fiber-like formation and (B) pore-like formation  

  • In the text, authors should specify the solvent (water?) for which SASA was computed.Based on the results in Figure (3) the authors state that the gyration radius and the SASA are decisive in the aggregation process. Actually, the language used (“driven forces”) appears not correct, and in the text there is no clear interpretation of the two quantities. Furthermore, the two quantities have been used in the literature (see for example Yu and Schatz J. Phys. Chem. B 2013, 117, 9004-9013) and I think that adding a few references on this topic can help the reader understand the relevance of the two quantities in the aggregation process.

Author reply: Thanks for the constructive suggestion. To investigate the aggregation process more deep, the solvent-accessible surface area (SASA) of Aβ peptides in DPPC membrane was calculated. The correlations between Aβ peptides and DPPC membrane is provided by examining SASA and the radius of gyration (Rg) of Aβ peptides. The free energy landscape is determined by calculating the normalized probability from a possibility distribution of SASA and Rg as seen in Equ.1 and Equ.2, where X and Y represents the SASA and Rg. To explain the SASA and Rg more clearly, we have implemented a few references on this topic. The language (“driven forces”) have changed as follows: As mentioned in Yu’s paper, the change of structure accompanied with the change of SASA and Rg. It could greatly change the free energy with the change of enthalpy and entropy, and the Aβ peptides could aggregate into pore-shaped structure or fiber-like structure. We have revised this part in manuscript with red color.

  • Please indicate the unit for density in Figure 6.

Author reply: Thanks. The number of density of molecules were calculated with the number of molecules in the volume of section along the z axis of system. We have indicated the unit of number of density in Figure 6.

Figure 6. the density of water molecules (black line) and Na+ ions (red line) along z axis of DPPC membrane.

  • Conclusion should be improved. What is the relevance of conclusions drawn from only one structure of the pore?

Author reply: As reviewer mentioned, the conclusion from only one structure of the pore is not convincing. The experiment of Natàlia Carulla and coworkers has confirmed that the pore formation of Aβ(1-42) oligomers [R1. S. Ciudad, E. Puig, T. Botzanowski, M. Meigooni, A. S. Arango, J. Do, M. Maxim, B. Mariam, C. Stephane, M. Giovanni, C. Sarah, O. Vladislav, T. Emad, B. Benjamin, and Carulla, N. Nat. Commun., 2020, 11, 1-14.]. The experiment could partly confirm the phenomenon of pore formation with Aβ peptides, the results of the simulation is convincing. Thus, the mechanism of pore formation of Aβ peptides and the water permeation of aggregated pore was explored. In further, we will also study other pore structures and mechanisms according to different polypeptide compositions and different membranes.

We have revised the conclusion as follows: The free energy landscape were measured as a function of the change of Rg and SASA, and the pore-like structure is more stable than fiber-like structure on the membrane. The aggregates do not consist of a specific number of peptides on the membrane, but rather they grow in size over time. In addition, membrane lipid composition also has the effect on the aggregation process of Aβ peptides. Herein, the pore formation of Aβ peptides on membrane was observed, and water permeation of Aβ channel was investigated by AA simulation. Our results demonstrate that the Aβ channel formed by Aβ peptides on the DPPC membrane has a good water permeation.  

Reviewer 2 Report

Reviewer Recommendation and Comments for the manuscript "Pore formation mechanism of A-beta peptide on the fluid membrane: A combined coarse-grained and all-atomic models"

The authors show a combination of coarse-grained and all-atom simulations to model the formation of A-beta amyloid pores (or ionic channels) and the aggregation of these A-beta amyloids.

My comments are: 

I would like to know if the authors have verified that their results come out in the same way with the new versions of the MARTINI v3 force field (August 2021) and the new versions of GROMACS (the one they use is from 2013, and there have been new optimisations and corrections in some cases in the current versions).

How many trajectories did the authors launch to obtain the conclusions in lines 127 and below?

The readers could feel that the authors fall into contradictions. In line 124 authors said that the number of clusters remains at four from 500ns until the end of the simulation, however, in line 133 they said that the number of clusters is reduced to 3.

The authors only show a brief part of the simulation in figure 4. I think it would be more interesting if the authors showed the analysis for the whole simulation, and to review what is shown in that figure, they zoomed in on the part in the figure, that they wants to focus.

The authors select the last martini model structure to generate the AA model. I think it would have been more correct to select several structures of the trajectory in the CG model and launch several trajectories that would more correctly be mapping the phase space.

typo in line 46: write at al, instead of et al

typo in line 88, 5us instead of 5 picoseconds

typo in line 160, 5us instead of 5 picoseconds

Author Response

Reviewer 2:

Comments:

  • I would like to know if the authors have verified that their results come out in the same way with the new versions of the MARTINI v3 force field (August 2021) and the new versions of GROMACS (the one they use is from 2013, and there have been new optimisations and corrections in some cases in the current versions).

Author reply: Thank you. We have not compared with the new versions of the MARTINI v3 force field (August 2021) and the new versions of GROMACS. The experiment of Natàlia Carulla and coworkers has confirmed that the pore formation of Aβ(1-42) oligomers [R1. S. Ciudad, E. Puig, T. Botzanowski, M. Meigooni, A. S. Arango, J. Do, M. Maxim, B. Mariam, C. Stephane, M. Giovanni, C. Sarah, O. Vladislav, T. Emad, B. Benjamin, and Carulla, N. Nat. Commun., 2020, 11, 1-14.]. The experiment could partly confirm the phenomenon of pore formation with Aβ peptides, the results of the simulation is convincing. Based on this, the mechanism of pore formation of Aβ peptides and the water permeation of aggregated pore was explored. In further, we will also study the aggregation mechanism of Aβ peptides based on the new versions of MARTINI v3 force field and GROMACS software.

  • How many trajectories did the authors launch to obtain the conclusions in lines 127 and below?

Author reply: After 500ns simulation, the number of clusters remains 4 until the end of the 5μs simulation. The simulation was repeated 3 times, and the number of clusters remain 3 and 3 in other two trajectories. We have revised this part into the manuscript.

  • The readers could feel that the authors fall into contradictions. In line 124 authors said that the number of clusters remains at four from 500ns until the end of the simulation, however, in line 133 they said that the number of clusters is reduced to 3.

Author reply: Thanks for the nice suggestion. It is our fault to describe the process of change of the number of cluster. In the 4000ns, the number of cluster is still 4, and the number of pore shaped clusters increased to 3. It shows more fiber-like structure of aggregated Aβ peptides have changed into pore-like structure. We have revised this part in manuscript in red.

  • The authors only show a brief part of the simulation in figure 4. I think it would be more interesting if the authors showed the analysis for the whole simulation, and to review what is shown in that figure, they zoomed in on the part in the figure, that they wants to focus.

Author reply: Thanks for the constructive suggestion. The order parameter (Sorder) of the largest Aβ peptides cluster in two systems on different membrane were measured as as function of simulation time. The alpha C atoms of Aβ channel in the largest cluster were used to analysis. As you seen in Figure 4, the S value of Aβ channel on two different membrane decreased in the first stage with the aggregation of Aβ peptides. However, the S value of Aβ channel on DPPC membrane is much higher than that of Aβ channel on the POPC membrane. It means that the arrangement of Aβ peptides in the Aβ channel on the DPPC membrane are more parallel than that on the POPC membrane. It is also the reason that the pore formed by Aβ peptides on DPPC membrane are larger than that formed by Aβ peptides on POPC membrane. Due to the membrane fluidity, the S value of Aβ channels from two systems fluctuated a lot in the simulation. It facilitated the conformation change of Aβ peptides in the aggregation process.

Figure 4. The order parameter of the alpha C atoms of the largest Aβ channel as a function of time on two lipids membrane: POPC and DPPC.

  • The authors select the last martini model structure to generate the AA model. I think it would have been more correct to select several structures of the trajectory in the CG model and launch several trajectories that would more correctly be mapping the phase space.

Author reply: As the reviewer mentioned, To select several structures of the trajectory from the CG model to transform the all-atomic model and launch several trajectories is much better for the accuracy of the results of simulation. The phenomenon of the pore formation of Aβ peptides from simulatioon is confirmed by the experiment of Natàlia Carulla and coworkers [R1. S. Ciudad, E. Puig, T. Botzanowski, M. Meigooni, A. S. Arango, J. Do, M. Maxim, B. Mariam, C. Stephane, M. Giovanni, C. Sarah, O. Vladislav, T. Emad, B. Benjamin, and Carulla, N. Nat. Commun., 2020, 11, 1-14.]. Thus, the pore formation structure of Aβ peptides from the end of CG simulation ws selected to transfer into all-atomic model, the mechanism of water permeation of Aβ channel based on this structure could be convinced. We have revised this part in the revised manuscript in red.  

typo in line 46: write at al, instead of et al

typo in line 88, 5us instead of 5 picoseconds

typo in line 160, 5us instead of 5 picoseconds

Author reply: Thank you,we have revised the typos and syntax errors through the full manuscript.  

Round 2

Reviewer 1 Report

The second version of the manuscript is improved after changes introduced by the authors. However, I suggest to check English language (lines 125, 148, 215, 263).

Reviewer 2 Report

The first thing is to thank the authors for the changes they have made to the article.

I understand the comments of the authors about the versions of the programs and codes used. However, I hope that they understand my recommendation that they review the new modifications for future research on the same topic.

Taking this into account, I recommend the publication of the manuscript.